# An Autonomous Control Framework of Unmanned Helicopter Operations for Low-Altitude Flight in Mountainous Terrains

**Zibo Jin [1], Lu Nie [2], Daochun Li [1], Zhan Tu [1,3,*] and Jinwu Xiang [1]**

[1] School of Aeronautic Science and Engineering, Beihang University, Beijing 100191, China; jinzibo@buaa.edu.cn (Z.J.); lidc@buaa.edu.cn (D.L.); xiangjw@buaa.edu.cn (J.X.)
[2] Beijing Institute of Space Long March Vehicle, Beijing 100076, China; nielu@buaa.edu.cn
[3] Institute of Unmanned System, Beihang University, Beijing 100191, China
[*] Correspondence: zhantu@buaa.edu.cn; Tel.: +86-(010)-82338079

**Abstract:** Low-altitude flight in mountainous terrains is a difficult flight task applied in both military and civilian fields. The helicopter has to maintain low altitude to realize search and rescue, reconnaissance, penetration, and strike operations. It contains complex environment perception, multilevel decision making, and multi-objective flight control; thus, flight is currently mainly conducted by human pilots. In this work, a control framework is implemented to realize autonomous flight for unmanned helicopter operations in an unknown mountainous environment. The identification of targets and threats is introduced using a deep neural network. A 3D vector field histogram method is adopted for local terrain avoidance based on airborne Lidar sensors. In particular, we propose an intuitive direct-viewing method to judge and change the visibilities of the helicopter. On this basis, a finite state machine is built for decision making of the autonomous flight. A highly realistic simulation environment is established to verify the proposed control framework. The simulation results demonstrate that the helicopter can autonomously complete flight missions including a fast approach, threat avoidance, cover concealment, and circuitous flight operations similar to human pilots. The proposed control framework provides an effective solution for complex flight tasks and expands the flight control technologies for high-level unmanned helicopter operations.

**Keywords:** autonomous flight control; unmanned helicopter operation; terrain avoidance; visual servo control; threat avoidance

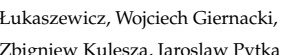

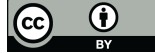

## 1. Introduction

Unmanned aerial vehicles (UAVs) have received substantial interest from the research community and the general public alike in recent years [1,2], especially small UAVs and multi-rotors, whose low cost and convenient use provide ideal testbeds and development impetus for innovative technologies of control approaches [3,4], advanced intelligent perception [5], and complete autonomy [6–9]. Small and medium UAVs can hardly meet the demands of high payloads and long flight distances. Therefore, more and more large-scale UAVs are being developed to provide long-endurance flights and perform various missions like manned aircraft. There are also related research projects developing independent autonomous equipment [10,11] or executing modifications [12] to convert manned aircraft to UAVs. There is a great need to investigate autonomy flight technologies for application scenarios of large-scale UAVs, manned aircraft, and helicopters for unmanned operations.

Helicopters demonstrate unique characteristics of maneuverability and low-speed performance, significantly extending their application in both military and civil fields [13]. With the development of advanced sensing devices and technologies, various sensor systems including cameras, radar, laser/light detection and ranging (Lidar), electro-optical (EO) system, acoustic system, and infrared (IR) sensors are deployed on helicopters to

realize a higher level of perception and automation [14,15]. Researchers have been motivated to investigate autopilot technologies based on multi-source information perception to complete typical helicopter missions, such as target tracking [16–18], autonomous landing [16,19,20], and obstacle avoidance [21–24]. There have also been some studies of unmanned helicopter operations focused on specific flight scenes and missions. Gimenez [25,26] presented a transportation system for carrying a suspension payload through two helicopters considering collision avoidance, wind disturbance, and reasonable distribution of load weight. A ship–helicopter cooperative system was extensively investigated to allow helicopters to automatically approach and land on vessel decks [27–30]. Many studies demonstrated the landing performance of helicopter recovery on the vessels under visual guidance [31,32]. Chen [33] presented an efficient algorithm for the path-planning problem of multiple-helicopter formations in a realistic environment. Different solutions and control frameworks for formation flight and formation reconfiguration can be found in [34–36]. Recently, some new control frameworks including deep reinforcement learning framework [37,38] and genetic fuzzy trees [39] were applied and achieved good results. Chamberlain [40] presented an autonomy package allowing a full-scale unmanned helicopter to automatically fly through unmapped, obstacle-laden terrain, find a landing zone, and perform a safe landing near a casualty, all with no human control or input. Nikolayevich [41] proposed a new method based on enhanced 3D motion primitives for 3D path planning close to the flight dynamics limits of helicopters, enhancing their assistance and autonomy in missions. Schopferer [42] studied onboard and online flight path planning for small-scale unmanned rotorcraft to plan safe, dynamically feasible, and time-efficient flight paths using cubic splines. To summarize, the existing studies of unmanned helicopter operations mainly focused on basic tasks such as path planning, target tracking, and obstacle avoidance. Due to the lack of a higher level of autonomy and integrated control frameworks, few adequate solutions of complex tasks with diversified flight missions have been proposed.

Low-altitude flight in complex mountainous terrains is a difficult flight task applied in many fields. Especially for military applications, a low-altitude penetration flight is a typical example making use of the ultralow-altitude maneuvering of helicopters, so as to effectively use the terrain to avoid the detection and threat of the defense system, as well as improve flight survivability. Low-altitude flight is also widely used in civil fields for low-level reconnaissance, remote site material delivery, search and rescue, and casualty evacuation. Matthew [43] studied the low-altitude flight of a full-scale helicopter in complex terrains and demonstrated a tight integration of terrain avoidance, control, and autonomous landing. During the low-altitude flight, increasing levels of concealment are achieved by adopting different tactics such as terrain following, terrain avoidance, and threat avoidance [44]. For low-altitude flight, obstacle avoidance and terrain avoidance are the core flight tasks [45]. Zheng [46] designed a real-time flight control algorithm combining the fuzzy obstacle avoidance algorithm with the L1 control algorithm for helicopter low-altitude flight in complex environment. Chandrasekaran [47] reviewed helicopter wire strike protection and prevention devices for low-altitude flight, and carried out a multicriteria decision-making analysis to rank different wire strike prevention methods. Merz [48] introduced a system enabling robotic helicopters to fly safely without user interaction at low altitude over unknown terrain with static obstacles. Wang [49] proposed a collision avoidance strategy method and the corresponding calculation approach of optimal collision avoidance for small unmanned helicopters in low-altitude applications. There have also been many studies focusing on UAV obstacle avoidance in dynamic building environments [50,51]. Low-altitude flight also involves threat/target identification, visibility judgement, multilevel decision making, and multi-objective flight control. It poses a challenge to the implementation of unmanned operations. To automatically realize unmanned low-altitude flight like human pilots. It is necessary to establish an integrated control framework, which was rarely studied systematically in previous research.

In this research, the unmanned helicopter operations of low-altitude flight in complex mountainous terrains are investigated in detail. The flight scenes and mission requirements of low-altitude flight are discussed considering target/threat recognition and decision making on the basis of the recognition result. Low-altitude flight is divided into several basic tasks: target/threat recognition, target tracking, threat avoidance, and terrain following. We developed the control methods of the basic tasks according to the application scenarios of the mountainous terrains. A new method for judging target visibilities is proposed, which is inspired by the human perceptual method and is especially suitable for a complex and unknown environment. Then, we combined the basic tasks and established a coupled control method to realize flight missions including active approaching, threat avoidance, and circuitous flight operations. The control methods presented in this research were verified through high-realistic flight simulations.

The remainder of the paper is organized as follows: we briefly describe the mission requirements of low-altitude flight in mountainous terrains and present the modeling method of the simulation environment in Section 2; the implementation of target recognition, visual servo control, and terrain avoidance is investigated in Section 3, which also contains simulation verification and performance evaluation; Section 4 details the visibility judgment method and proposes the overall control framework based on a finite state machine; Section 5 presents the simulation validation of the proposed control framework; lastly, the conclusions are summarized in Section 6.

The main contributions and innovations of this research are listed as follows:

- We extend the low-altitude penetration flight by introducing target recognition and threat determination into the existing tactics [52–54], providing a wide perspective for the complicated flight tasks and various flight scenes of helicopter low-altitude flight.
- The helicopter visibility with respect to ground threats or specific facilities is investigated in this research, which was rarely studied in previous studies about threat avoidance or survivability assessment [55,56]. We also propose a direct viewing method to judge and change the visibility quickly and robustly.
- On the basis of the visibility judgement, an integrated control framework is established using the finite state machine. Compared with many existing studies [13,19,40,47,57], this framework focuses on solving complex multi-objective flight tasks and realizing unmanned helicopter operations of cover concealment and circuitous flight similar to human pilots.

## 2. Problem Formulation

### 2.1. Low-Altitude Flight in Complex Mountainous Terrains

Low-altitude flight is generally applied for mountainous or undulating terrains, where helicopters can make full use of the terrain to block detection and give full play to the advantages of mobility. Here, the low-altitude penetration flight used in military fields provides a great example which covers massive flight scenes of the low-altitude flight. We intend to state the basic tasks of the penetration flight and extend them to develop a more comprehensive control framework for the low-altitude flight. The main feature of penetration flight is maintaining a low flight altitude to avoid radar detection, and flying covertly to avoid ground defense and various detectors. To increase the level of concealment, TF/TA$^2$ tactics [58] have been developed and adopted as typical flight tasks of the penetration flight. The tactics are described as follows:

- Terrain following: flight maneuvering with the terrain contour in the vertical plane according to the predetermined minimum ground clearance. This penetration method can use terrain cover and reach the destination in a short time.
- Terrain avoidance: flight maneuvering in the azimuth plane, flying around mountains and other tall obstacles. This penetration method can make full use of the terrain as cover and facilitate hiding, but increases the likelihood of colliding with terrain obstacles.

- Threat avoidance: flight maneuvering in the azimuth plane, avoiding detection and weapon attacks, fully approaching the target, realizing sudden attacks, and reducing enemy interference.

The above tactics constitute the basic needs of the penetration flight, but there is still a lack of an integrated decision-making framework or specific maneuvering methods when facing various flight missions. Therefore, we extend the above tactics, and put forward broader flight tasks as follows:

- Target/threat recognition: identifying the target/threat facilities during the flight and determining the threat degrees; making maneuvering decisions on the basis of the recognition result.
- Target approaching: identifying the target using airborne cameras, tracking and approaching the target through visual servo control, avoiding terrain obstacles, and maintaining the ability to approach the target when it is blocked or temporarily lost.
- Cover concealment: when a threat is detected, finding cover through the terrains and moving to the terrain cover to escape the threat; discriminating and changing the helicopter's visibility through flight maneuvers.
- Circuitous flight: comprehensive flight maneuvering around the terrain, avoiding the threat, and following the terrain contour near the predetermined heading, so as to finally reach the destination safely.

In this work, we focus on the implementations of the unmanned helicopter operations of these flight tasks. Furthermore, we build a decision-making framework which can autonomously deal with different flight tasks without human intervention.

### 2.2. Modeling Method of the Simulation Environment

A helicopter exhibits six-degree-of-freedom rigid-body dynamics. The flight dynamic equations are as follows:

$$\dot{V} = \frac{F}{m} - \Omega V, \tag{1}$$

$$\dot{S} = I^{-1}M - I^{-1}\Omega IS, \tag{2}$$

$$\dot{\alpha} = ES, \tag{3}$$

$$\dot{P} = R_{BG}V, \tag{4}$$

where $V = [u\ v\ w]^T$ is the linear velocity, $S = [p\ q\ r]^T$ is the angular velocity, $\alpha = [\varphi\ \theta\ \psi]^T$ is the Euler angle of roll, pitch, and yaw, $P = [X\ Y\ Z]^T$ is the position vector in ground coordinates, $m$ is the mass of the helicopter, and $F$ and $M$ are the forces and moments of the components of the helicopter.

$I$ is the moment of the helicopter inertial matrix, $\Omega$ is the angular rate antisymmetric matrix, $R_{BG}$ is the conversion matrix from body coordinates to ground coordinates, and $E$ is the conversion matrix from body angular velocity to Euler angular velocity.

$$R_{BG} = \begin{bmatrix} \cos\theta\cos\psi & \sin\theta\sin\psi\cos\psi - \cos\varphi\sin\psi & \sin\theta\cos\varphi\cos\psi + \sin\varphi\sin\psi \\ \cos\theta\cos\psi & \sin\theta\sin\varphi\sin\psi + \cos\varphi\cos\psi & \sin\theta\cos\varphi\cos\psi - \sin\varphi\sin\psi \\ -\sin\theta & \sin\varphi\cos\theta & \cos\varphi\cos\theta \end{bmatrix}. \tag{5}$$

$$E = \begin{bmatrix} 1 & \sin\varphi\tan\theta & \cos\varphi\tan\theta \\ 0 & \cos\varphi & -\sin\varphi \\ 0 & \sin\varphi/\cos\theta & \cos\varphi/\cos\theta \end{bmatrix}. \tag{6}$$

For the helicopter dynamic model, $F$ and $M$ are generated by the aerodynamic forces of the fuselage and the control forces which originate from the main rotor thrust and tail rotor thrust. The helicopter has large aerodynamic interference and is a highly coupled complex dynamic system, which makes it difficult to establish a fully dynamic model. Therefore, the linearized dynamic model was adopted in this research to simulate the

helicopter dynamic responses. The linearized model is obtained through frequency-domain identification of the flight experiment, and it is widely used in helicopter controller design and dynamic characteristic analysis. The complete linearized dynamic model can be illustrated in state-space representation as follows [59]:

$$\dot{x} = A_m x + B_m u, \tag{7}$$

where $A_m$ and $B_m$ are the system matrix and control matrix at different equilibrium points, establishing a linear-parameter-varying (LPV) helicopter dynamics model. $x = [u\, w\, q\, \theta\, v\, p\, \phi\, r\, \psi]$ is the state vector and $u = [\delta_e \delta_c \delta_a \delta_p]$ is the control input vector, where $\delta_c$ is the collective control input of the main rotor blade, $\delta_e$ and $\delta_a$ are the cyclic control inputs giving the explicit pitch in longitude and lateral directions, and $\delta_p$ is the collective pitch for the tail rotor.

　　Cascade PID (proportion integration differentiation) controllers were adopted to realize the low-level control of the helicopter. As shown in Figure 1, the helicopter control system was divided into the longitude channel, lateral channel, altitude channel, and yaw channel. For each channel, an independent cascade PID controller was applied. The inner loop controllers maintain the attitude stability, while the middle loop and outer loop controllers are used to track speed or position commands. The stability analysis and convergence of the cascade PID framework are essential for the controller implementation, but beyond the scope of this work. A detailed analysis can be found in [51] for further discussion.

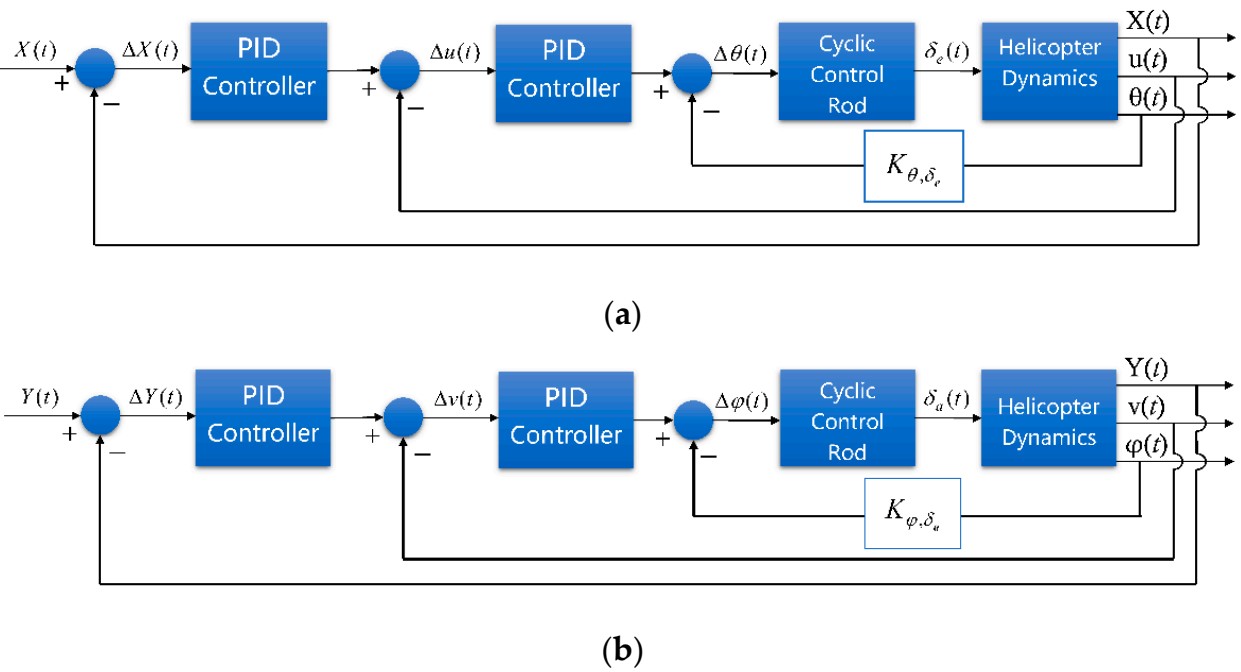

(a)

(b)

**Figure 1.** *Cont.*

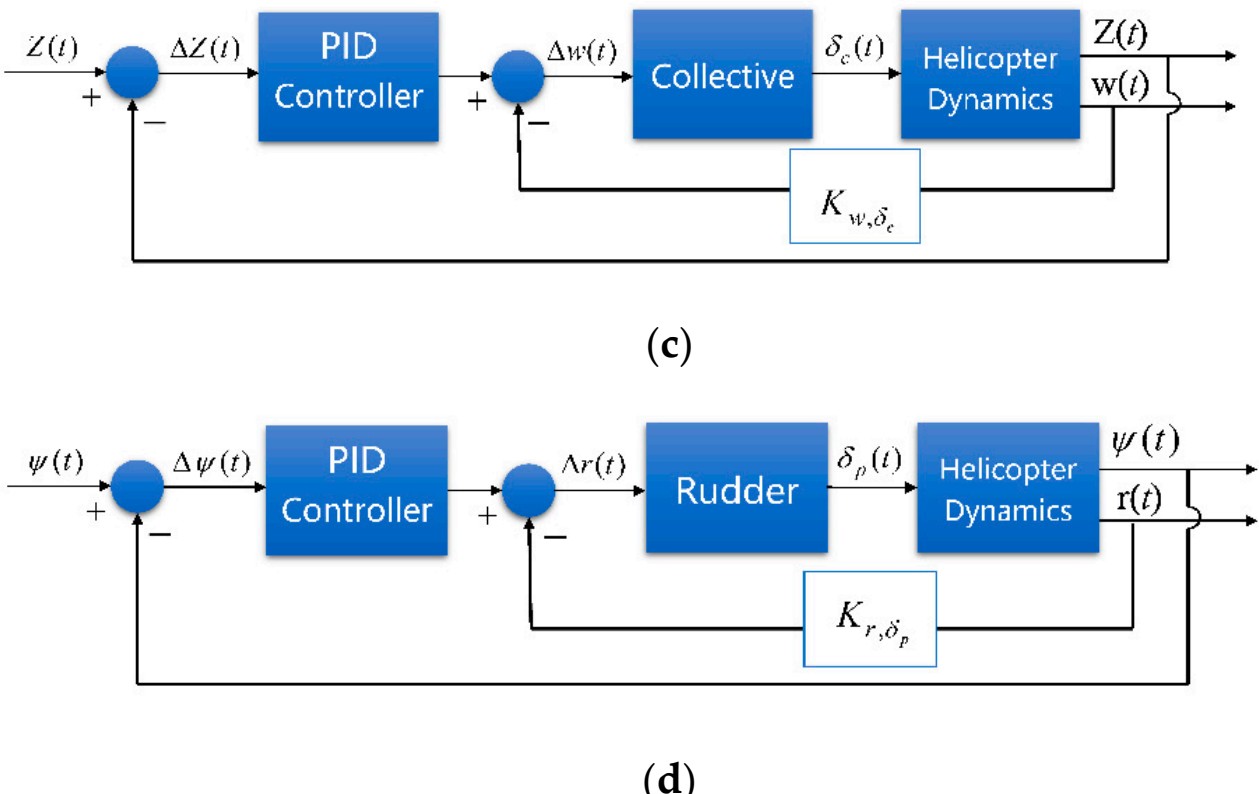

**Figure 1.** Cascade PID controllers of the helicopter control channels: (**a**) longitude channel; (**b**) lateral channel; (**c**) altitude channel; (**d**) yaw channel.

It should be mentioned that helicopters have multiple flight mode transitions arising from the complicated aerodynamic nature of thrust generation, while the control channels are strongly coupled. Various flight control methodologies have been developed for the flight control system of helicopters to improve the flight performance, which is beyond the scope of this research. We used simple and decoupling low-level controllers to clearly explain the implementation of visual servo control and terrain avoidance. In addition, the simple controllers used in this study would cause more oscillations and longer adjustment time during the flight. This required a high-level control framework to provide more margins in the design stage, forcing the control framework to be more adaptive and practical for engineering realizations.

A simulation based on realistic scenarios is a crucial part of testing algorithms. We built the simulation environment using the Unreal Engine package to visualize scenarios with realistic graphics and generate sensor data. A co-simulation framework was used to realize the communication interface between Unreal Engine and MATLAB Simulink, as shown in Figure 2. For each simulation step, the helicopter dynamic model received the flight control signals and updated the flight state. The flight state was sent to the simulation environment through a communication interface, driving the Unreal Engine to realize real-time virtual rendering. Various sensors were modeled to obtain ground-truth data in the simulation environment. The established control framework receives the sensor data and output control signals to the helicopter model, forming a closed-loop simulation system.

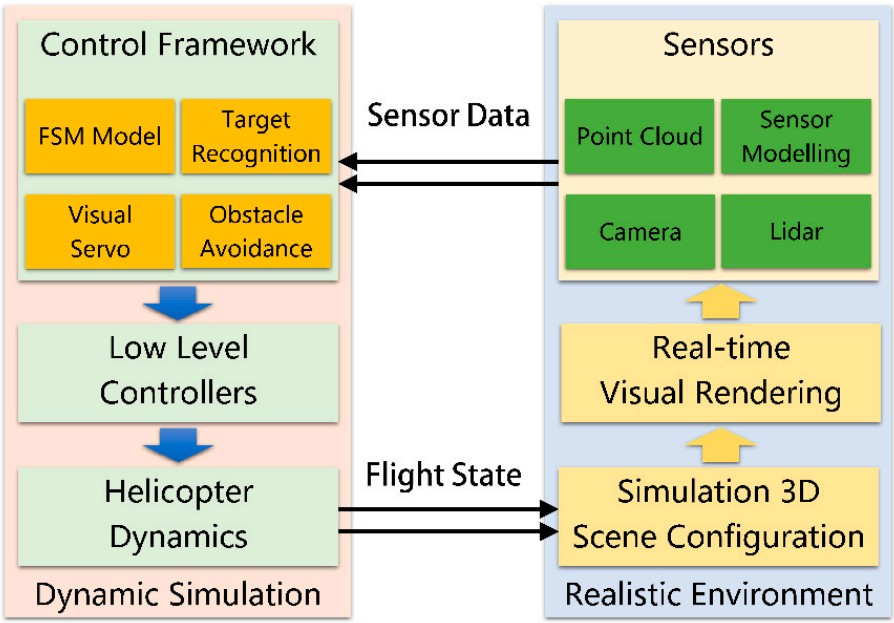

**Figure 2.** Closed-loop system of the realistic simulation environment.

A realistic scenario based on a mountainous map was built to carry out flight simulations of the low-altitude flight, as shown in Figure 3. We set up natural terrains and different facilities in the scenario. The terrains and facilities were built using static meshes, which could be detected by the virtual Lidar sensors of the helicopter. The virtual camera was mounted around the helicopter to obtain visual information. Benefitting from the powerful lighting, rendering, and mapping ability of Unreal Engine package, the virtual camera was able to display real-scene lighting effects such as area shadows and diffuse reflection, providing a high-fidelity simulation environment for this research.

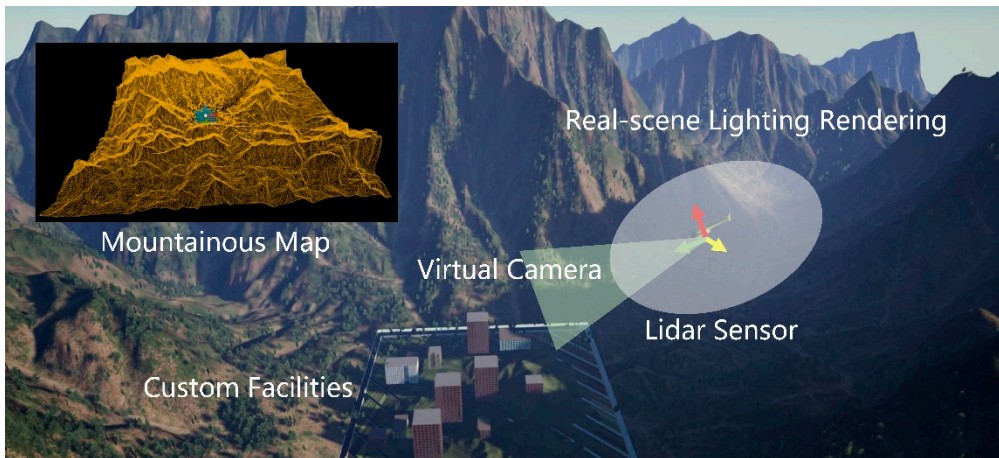

**Figure 3.** Realistic scenario of the mountainous terrain.

### 3. Target Tracking and Terrain Avoidance

*3.1. Target Tracking*

3.1.1. Target Recognition

For a low-altitude flight, the flight map information and accurate facility positions are generally unknown. Detecting targets and evaluating threat degrees represent the basis of decision making in unmanned helicopter operations. We expect that the target recognition method should be able to cover different types of targets as best as possible to deal with various unknown facilities that may appear in mountainous terrains. In this research,

target recognition was realized using the YOLOv2 network, which can be trained offline on labeled images to cover a large number of target features [60]. The YOLO model runs a deep learning CNN (convolutional neural network) on an input image to decode the predictions and generate bounding boxes, as shown in Figure 4. The detection network contains a series of conventional, batch norm, and rectified linear unit (ReLU) layers. By labeling and adding training samples of specific scenes, the YOLO model can better recognize distant or fuzzy targets.

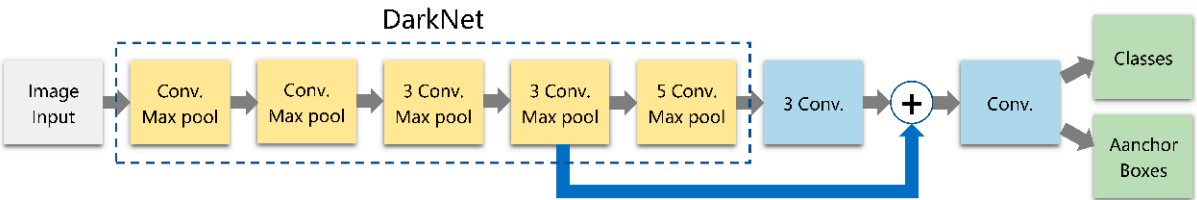

**Figure 4.** Structure of the YOLO Network.

　　The YOLO network introduces anchor boxes to improve the speed and efficiency for of detection. The anchor boxes are defined on the basis of object sizes in the training datasets. During detection, the predefined anchor boxes are tiled across the image. The position of an anchor box is determined by mapping the location of the network output back to the input image. The object detectors learn offsets to apply to each tiled anchor box, refining the anchor box position and size. The network predicts five coordinates for each bounding box: $t_x$, $t_y$, $t_h$, $t_w$, and $t_o$. The cell is offset from the top left corner of the image by $(c_x, c_y)$, and the bounding box prior has width and height $p_w$, $p_h$; the predictions can be drawn as follows:

$$b_x = s(t_x) + c_x, \tag{8}$$

$$b_y = \sigma(t_y) + c_y, \tag{9}$$

$$b_w = p_w e^{t_w}, \tag{10}$$

$$b_h = p_h e^{t_h}, \tag{11}$$

$$Pr(object) \times IOU(object, b) = s(t_o), \tag{12}$$

where, $b_x$, $b_y$, $b_h$, and $b_w$ are the box position and size parameters, $\sigma$ refers to the sigmoid function, and $\sigma(t_o)$ is the value of confidence after sigmoid transformation. For target recognition in low-altitude flights, the anchor boxes of the YOLO network provide the target dynamics within the sight ranges. The positions and sizes of anchor boxes also imply the relative position and attitude information of the target, which can provide the basis for the visual servo control.

　　The detection efficiency and generalization capabilities of the YOLO network depend on the number and diversity of training data. Without loss of generality, we extracted the images of some buildings and vehicles in the simulation environment as recognition targets. The images were manually calibrated to form a training dataset, and the network was trained using the SGDM (stochastic gradient descent with momentum) method. The precision of the trained detector at varying levels of recall is shown in Figure 5. The YOLO network can be replaced by frontier and stronger algorithms to obtain better recognition performance, but it was considered fairly effective for the overall control framework of this research.

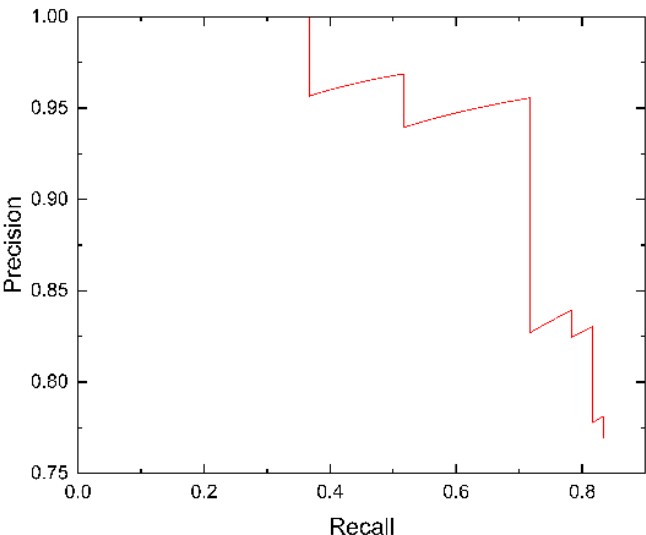

**Figure 5.** Precision–recall (PR) curve of the YOLO network.

### 3.1.2. Visual Servo Control

Visual servo control is an important part of unmanned helicopter operations in low-altitude flight. It can also help verify the stability of the YOLO detector and the effectiveness of the proposed helicopter controllers in this research. Therefore, we designed a typical flight task to carry out flight simulations on the basis of visual servo control. The helicopter identified the target through the YOLO detector, and then automatically hovered around the target, as shown in Figure 6. This has been used as the ground target tracking method of fixed-wing aircrafts [61,62]. For helicopters, we let the helicopter always head to the target, move horizontally through lateral maneuvers, and finally hover around the target.

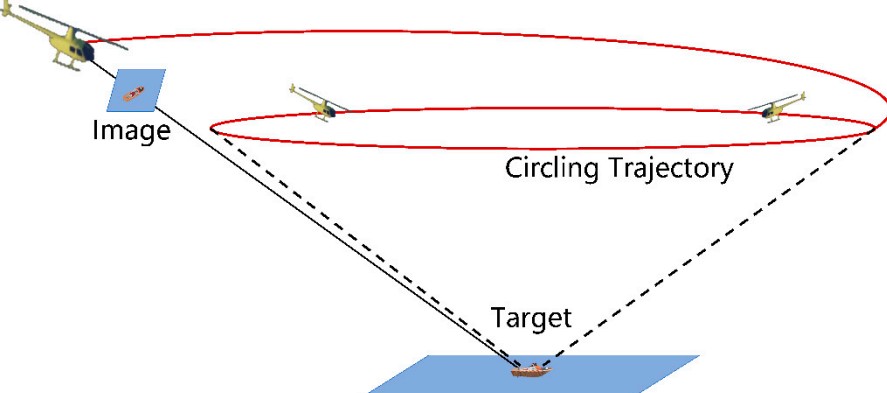

**Figure 6.** Circling flight around the target.

The commonly used methods of visual servo control can be divided into position-based visual servo (PBVS), image-based visual servo (IBVS), and end-to-end visual servo. PBVS establishes the mapping relationship between the image signal and the helicopter pose, calculates the pose information, and compares it with the required pose to form a closed-loop control. As PBVS needs accurate image signals, even a small error in the image measurements can lead to a large offset in the pose estimation. IBVS directly compares the image signal measured in real time with the image signal of required pose and uses the obtained image error for feedback control. The end-to-end servo method takes the captured image as the input, and directly outputs the control signals by constructing neural networks. However, there exist challenges including time complexity and servo stabilities when using this method. For a low-altitude flight, the target is generally unknown and

far away from the helicopter. It is difficult to extract a characteristic point of the image and calculate relative pose information using PBVS. Therefore, we adopted IBVS to realize visual servo control; the control framework is illustrated in Figure 7. We used the size and position of the anchor boxes provided by the YOLO network as the control commands of the helicopters control channels. The control commands were calculated as follows:

$$S_{box} = b_w b_h, \tag{13}$$

$$u_{Ref} = -k_u(S_{box} - S_{des}), \tag{14}$$

$$r_{Ref} = -k_r(b_x - b_{mid}), \tag{15}$$

$$v_{Ref} = V_{des} - \alpha_1(S_{box} - S_{des}) - \alpha_2(b_x - b_{mid}), \tag{16}$$

$$u_{Ref} = \begin{cases} u_{forward}, S^t = 0, b_y{}^{t-k} > 0 \\ u_{backward}, S^t = 0, b_y{}^{t-k} < 0 \end{cases}, \tag{17}$$

where $S_{box}$ represents the area of the anchor box, which characterizes the distance from the helicopter to the target. We defined a desired box area $S_{des}$ that points to the desired distance, and counted the error during feedback to the control command of linear velocity $u_{Ref}$. $k_u$ is the control gain. The target center is restricted to the horizontal center of the image to ensure that the helicopter is always oriented to the target. The error between target center $b_x$ and image center $b_{mid}$ is calculated and multiplied by the control gain $k_r$ as the control command of the yaw rate $r_{Ref}$. We set $V_{des}$ as the desired lateral velocity and correct it through the errors of box area and helicopter orientation to generate the control command of lateral velocity $v_{Ref}$, as shown in Equation (16), where $\alpha_1$ and $\alpha_2$ are the correction factors. In this way, priority is given to the helicopter maintaining its distance and orientation to the target, and then maneuvering laterally to hover around the target. Furthermore, in case the target is lost and deviates from the image, i.e., the anchor box area becomes 0, we preset longitudinal flight maneuvers to retrieve the target, as shown in Equation (17). When the target disappears from the top of the image, and the vertical coordinate of the target center before disappearing is positive ($b_y{}^{t-k} > 0$), the helicopter will fly forward to approach the target; otherwise, it will fly backward with linear velocity $u_{backward}$.

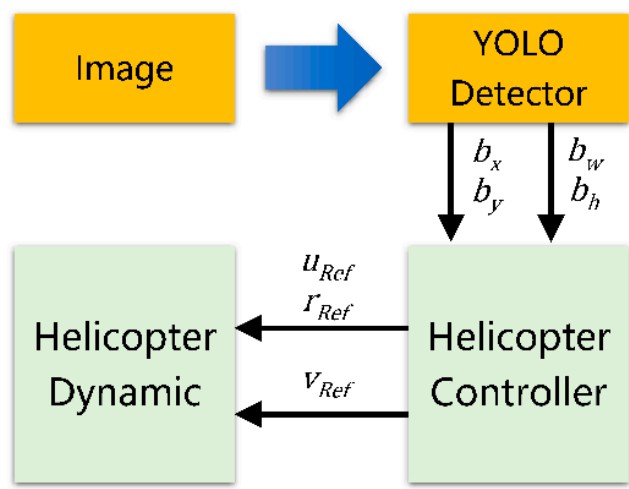

**Figure 7.** Visual servo control of the circling flight.

According to the above control frame, the IBVS method was established, and the control gains were determined. We set up the initial positions of helicopter and target, and we carried out flight simulations. The virtual camera was mounted under the helicopter body with a downward pitch angle of 20°. The sample time of the virtual camera was set as 0.01 s and the resolution of each frame was set to 640 × 360 to maintain image accuracy and detection efficiency. The flight path of the helicopter hovering around the static target is

shown in Figure 8. It can be seen that the helicopter approached the target from a distance and gradually maintained a stable circular trajectory. The linear velocity of the helicopter contained high-frequency oscillations caused by the visual servo control, but the overall trend was regular and stable, as shown in Figure 9. Figure 10 shows the anchor box size during the flight. The width and height of the anchor box changed periodically in a large range, which was caused by the different target poses under different viewing angles. The target was lost at some point and could be retrieved rapidly to continue tracking. On the basis of simulations around static targets, we carried out flight simulations to circle and hover around a moving target. The flight path is illustrated in Figure 11, which demonstrates the effectiveness of the control framework. Therefore, the proposed IBVS method can be further applied in low-altitude flight.

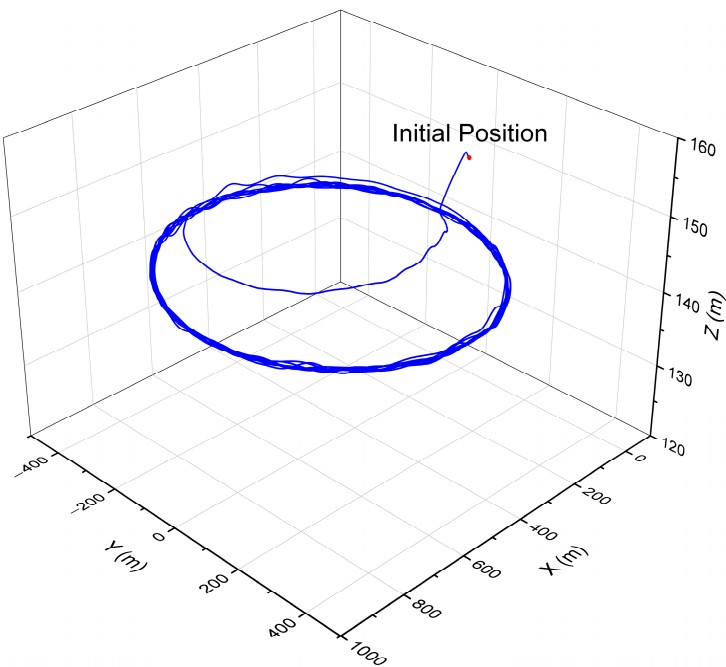

**Figure 8.** Flight path of the helicopter hovering around a static target.

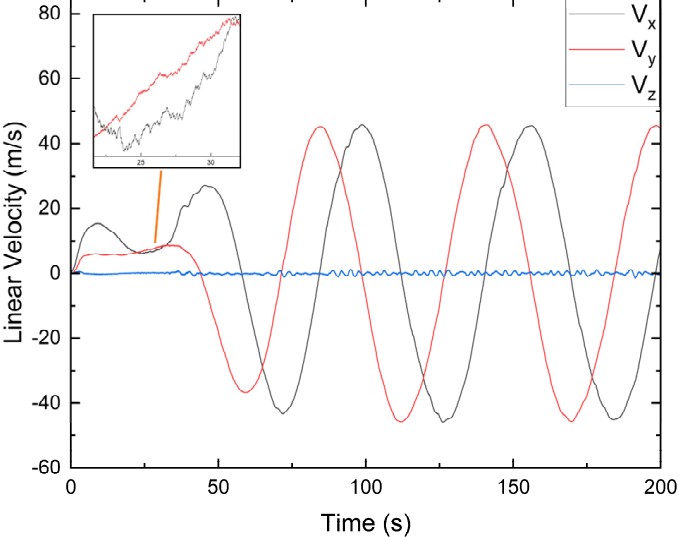

**Figure 9.** Helicopter linear velocities during a circling flight.

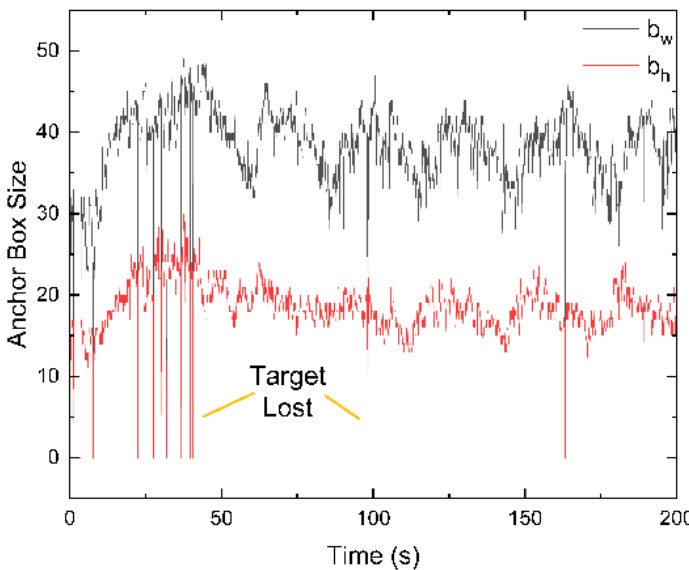

**Figure 10.** Anchor box size during a circling flight.

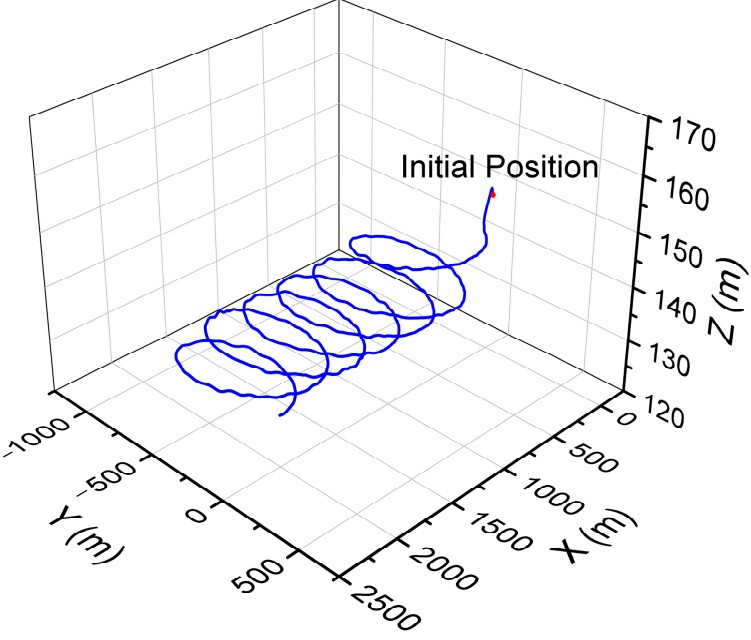

**Figure 11.** Flight path of the helicopter hovering around a moving target.

### 3.2. Terrain Avoidance

Terrain avoidance is a basic task in low-altitude flight, whereby the helicopter senses objects according to the airborne sensors, and autonomously executes obstacle avoidance and path planning in an unknown environment. Therefore, the global obstacle avoidance methods which rely on complete prior information are generally inappropriate. Moreover, low-altitude flight generally requires a helicopter to reach the destination rapidly to reduce flight time and risks. A reactive local obstacle avoidance method is more adaptable compared with a global mapping method. The terrains of mountainous areas are complicated and contain various environment objects. Radar and Lidar equipment have been commonly used for manned helicopters to sense and avoid terrain obstacles in mountainous terrains. Considering the sensing equipment, algorithm efficiency, spatial complexity, and application scenes, the 3D VFH (vector field histogram) method based on Lidar sen-

sors was established to realize terrain avoidance for unmanned helicopter operations in this research.

The 3D VFH algorithm originates from the widely used VFH algorithm in a 2D environment. It does not need specific map information, but it can provide multiple paths to maintain different requirements by designing different path weights, which is especially suitable for low-altitude flight. For 3D environment applications, the 3D VFH method divides the voxels near the helicopter into multiple cells through the two dimensions of the azimuth angle $\beta_z$ and the elevation angle $\beta_e$, as shown in Figure 12. The spherical voxels unfold into a 2D primary polar histogram, where each cell represents the possible direction of the helicopter.

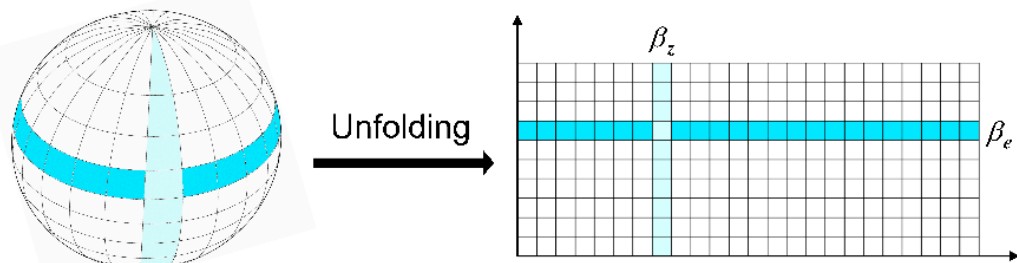

**Figure 12.** Formation of the 2D primary polar histogram.

We built a virtual Lidar sensor which was fixed under the helicopter cockpit in the simulation environment. It enabled the helicopter to sense the surrounding terrain obstacles in a determined range. The vertical and horizontal fields of view of the Lidar sensor were set as 60° and 360°, respectively. The detection range was set to 300 m to maintain the effectiveness and data scale. As shown in Figure 13, the Lidar sensor generated the point-cloud data in an ellipsoid range centered on the helicopter.

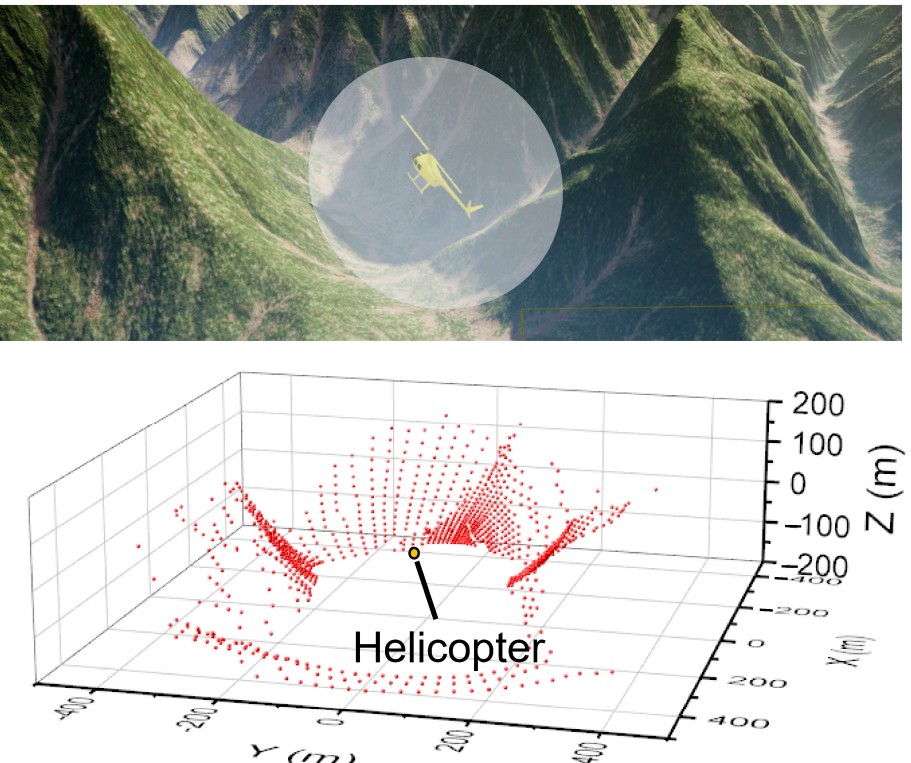

**Figure 13.** Generation of point-cloud data of virtual Lidar sensor.

For any node $P_i$ of the point-cloud data, we assume the coordinates $(x_i, y_i, z_i)$. The azimuth angle and elevation angle can be calculated using Equations (18) and (19), where $\alpha$ is the resolution of the 2D polar histogram, and the floor function creates natural numbers as the coordinates of the 2D primary polar histogram. Using the point-cloud data, we can evaluate the distance and size of terrain obstacles to calculate the risk weights, and add them to the 2D primary polar histogram. The weight of the voxels can be calculated using Equation (20), where $o_p$ is the occupancy certainty, $l_P$ is the Euclidean distance, which is influenced by the helicopter radius, safe radius, and voxel size, and $a$ and $b$ are predefined constant values. The detailed derivations can be found in [63].

$$\beta_z = \text{floor}(\frac{1}{\alpha}\arctan\frac{x_i}{y_i}). \tag{18}$$

$$\beta_e = \text{floor}(\frac{1}{\alpha}\arctan\frac{z_i}{\sqrt{x_i{}^2 + y_i{}^2}}). \tag{19}$$

$$H_{z,e} = \begin{cases} \sum_P o_p{}^2(a - bl_P) \text{, if } e \in \left[\beta_e - \frac{\lambda}{\alpha}, \beta_e + \frac{\lambda}{\alpha}\right] \text{ and } z \in \left[\beta_z - \frac{\lambda}{\alpha}, \beta_z + \frac{\lambda}{\alpha}\right] \\ 0, \text{ otherwise} \end{cases}. \tag{20}$$

The 2D primary polar histogram presents a simplified description of collision risks at different directions. A 2D binary polar histogram was established to further reduce the information. This was accomplished by comparing every cell in the 2D primary polar histogram with a threshold $\tau$. The size of the threshold depends on the helicopter radius, flight speed, sensor resolution, and bounding sphere size. When the cell weight is higher than $\tau$, the point will be 1 in the 2D binary polar histogram. When the value is lower than $\tau$, the point will be 0 in the 2D binary polar histogram.

The VFH method searches for available paths and detects openings by moving a window around the 2D binary polar histogram. This window marks the path passable if all the elements in the window are equal to 0. It defines three path weights combined for the candidate direction to select the path with lowest path weight $\mu$, as shown in Equation (20). The first path weight $\mu_1$ is used to multiply the difference between the target angle $k_t$ and the candidate direction $v_c$. The second path weight $\mu_2$ multiplies the difference between the helicopter yaw angle $\psi$ and the candidate direction $v_c$. The last path weight $\mu_3$ multiplies the difference between the previous selected direction $k_{t-1}$ and the candidate direction $v_c$. The function $\Delta(x, y)$ calculates the difference between the two direction vectors. By changing the path weights, multiple flight paths with different preferences can be obtained. For low-altitude flight in mountainous terrains, we can change the path weight allowing the helicopter to maintain low-altitude flight using turning motions or climbing motions to approach the target aggressively.

$$\mu = \mu_1 \cdot \Delta(k_t, v_c) + \mu_2 \cdot \Delta(v_\psi, v_c) + \mu_3 \cdot \Delta(k_{t-1}, v_c). \tag{21}$$

A narrow mountainous area with dense terrain obstacles was built to verify the terrain avoidance performance of the VFH method, as shown in Figure 14. We arranged an additional static mesh of obstacles in the terrains to increase the difficulty of obstacle avoidance. At each simulation step, the VFH method provided a desired direction and desired yaw angle according to the virtual Lidar sensor. For helicopters in low-altitude flight, turning maneuvers are more sensitive and stable than lateral maneuvers. Therefore, we took the desired yaw and vertical components of the desired direction as the helicopter control command to realize terrain avoidance in the low-altitude flight. The lateral control channel maintained the helicopter stability, and the longitudinal control channel changed the approaching speed of the helicopter to the destination.

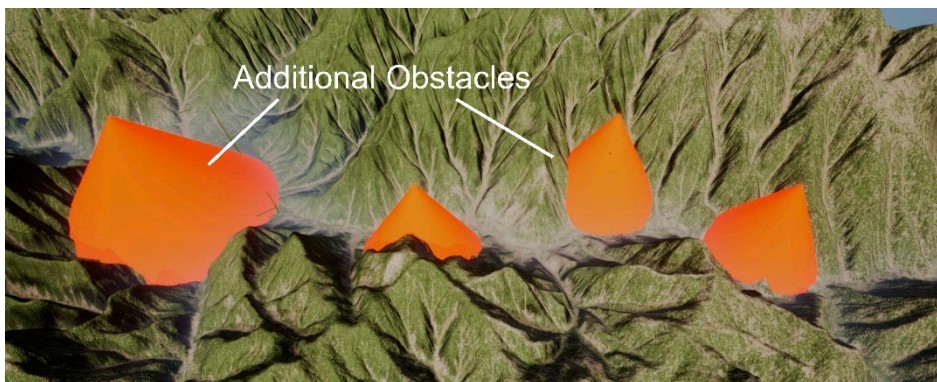

**Figure 14.** Narrow mountainous area with additional obstacles.

Lastly, we carried out flight simulations using the VFH method, where the helicopter was placed in a narrow mountainous area without map information. The virtual Lidar had a range resolution of 0.2 m. The vertical resolution and horizontal resolution of the Lidar were set as 2.5° and 5°, respectively. The vehicle size and minimum distance to obstacles of the VFH method were set as 5 m and 20 m. The sample time of the virtual Lidar was 0.1 s. A distant destination was defined, and the helicopter approached the destination and executed terrain avoidance during the flight. The path weights in Equation (21) were set as 3, 2, and 0.3, respectively. The flight path and linear velocities of the helicopter are illustrated in Figures 15 and 16, which demonstrates that the helicopter could maintain a low altitude and stably approach the destination. Figure 17 further shows the helicopter during the low-altitude flight. The helicopter avoided all the terrain obstacles in the narrow environment. This shows the good adaptability of the proposed VFH method in mountainous areas for the helicopter.

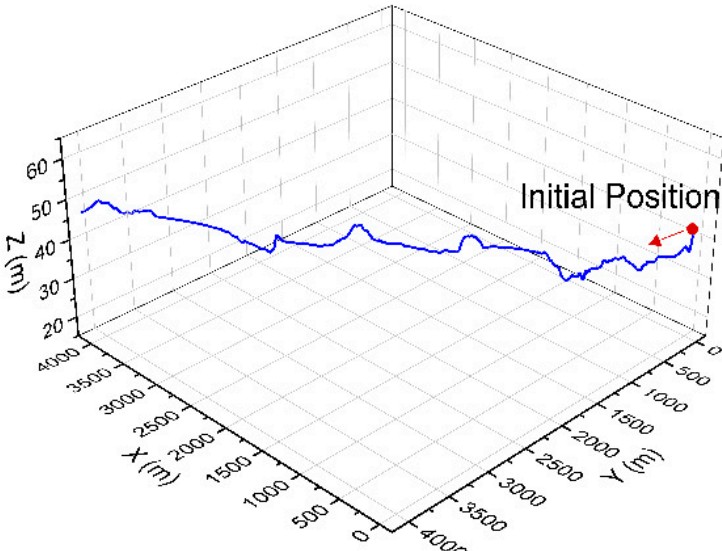

**Figure 15.** Flight path of the 3D VFH method.

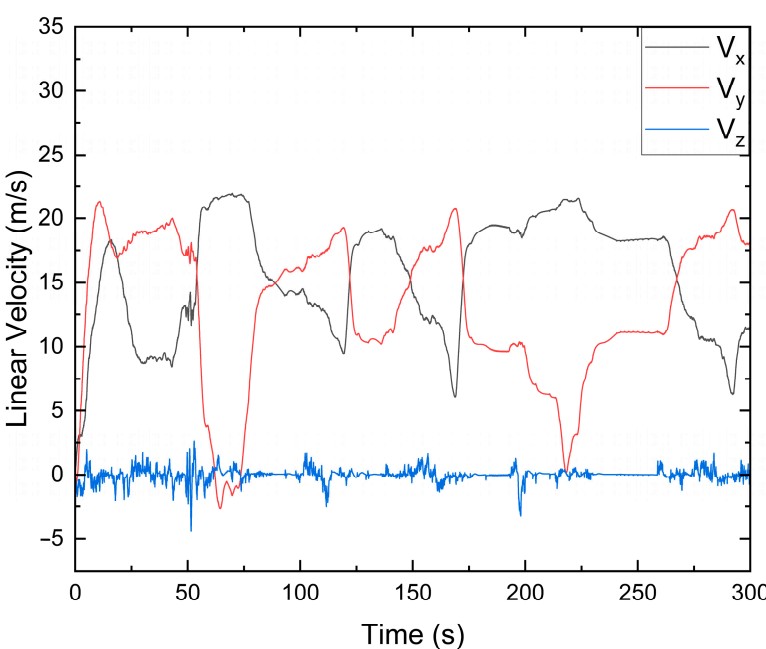

**Figure 16.** Helicopter linear velocities during the low-altitude flight.

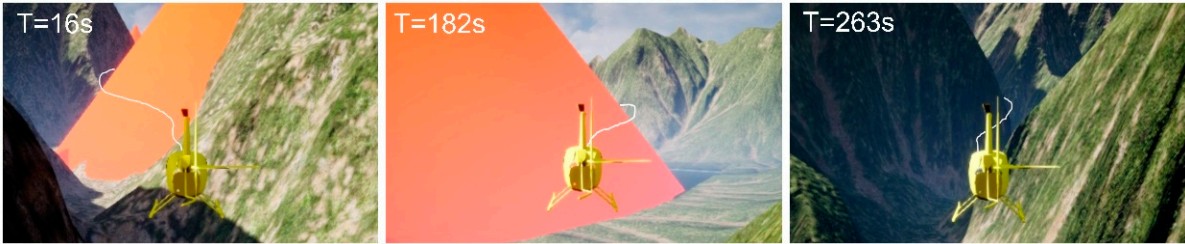

**Figure 17.** Helicopter during the low-altitude flight.

## 4. Autonomous Decision-Making Framework

### 4.1. Visibility Judgment

For low-altitude flight in mountainous terrains, helicopters can effectively avoid specific facilities or threats such as dense buildings, fires, and danger using obstacle avoidance and path replanning methods. In previous studies, these threats were often regarded as obstacles during the flight. However, in many situations of military applications, low-altitude flight requires the helicopter to resist ground detection and avoid ground defense to improve flight survivability. Since flying at low altitude can effectively block the detection of ground radars, the visibility of the helicopter to the ground facilities is crucial for decision making and maneuver selection. The helicopter needs to not only bypass and avoid the threats, but also escape from the sight range of the threats. Human pilots estimate the visibility of the helicopter using their eyes and intuition to make decisions such as taking cover, avoiding reconnaissance, or executing circuitous flight. On the other hand, the visibility judgement is quite difficult for unmanned operations, which generally needs complete state estimation, threat location, and accurate map information.

Here, we propose an intuitive direct-viewing method which can quickly judge the helicopter's visibility, as shown in Figure 18. The threat detection method uses a deep neural network, which is the same as the target recognition method presented in Section 3.1. We can add samples to the dataset and train a detection network to simultaneously identify targets and threats. When the helicopter detects a threat during low-altitude flight, it immediately turns to head to the target using the visual servo control method as mentioned in Section 3.1. At this moment, the helicopter is visible to the threat. To change its visibility,

the helicopter maneuvers laterally. This is because when the helicopter is heading to the threat, lateral maneuvers gain more variation than longitude and vertical maneuvers in the sight range of the threat. This also helps the helicopter to approach terrain cover in mountainous areas. During lateral maneuvers, if the threat is lost from view and the Lidar sensor can detect obstacles ahead, it can be considered that the line of sight between the helicopter and the threat is blocked, and the visibility is changed. In order to ensure the complete concealment of the whole helicopter, we set a margin to make the helicopter continue to fly laterally for a certain distance after its visibility changed.

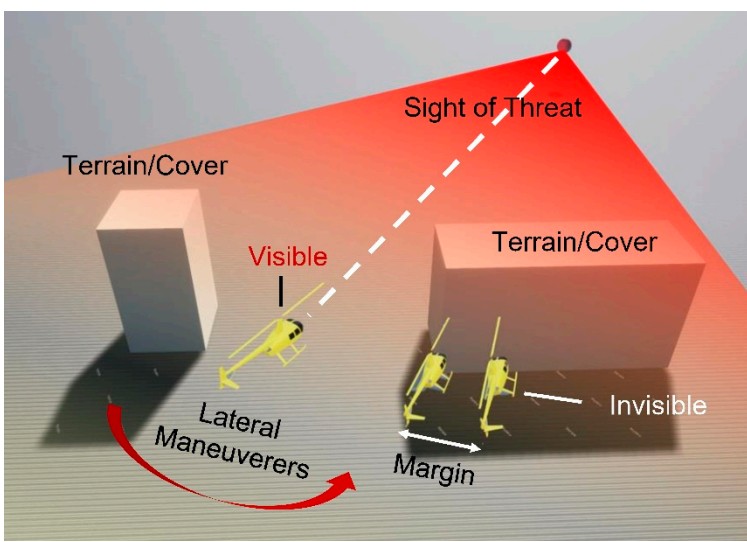

**Figure 18.** Direct-viewing method of visibility judgement.

The proposed method of the visibility judgment is essentially based on the line of sight, which is inspired by the perception method of human intuition. It is simple and effective, but requires the helicopter to keep heading to the threat. Obviously, continuously heading to the threat is not the optimal method to avoid it. This limits the helicopter's movement and possibly further exposes the helicopter to the threat. However, it provides a fast and reliable method to realize real-time judgement of the visibility without comprehensive map information or a complex calculation process. Moreover, the helicopter has good lateral maneuverability; when facing a threat, lateral maneuvers are faster and more continuous than turning or other maneuvers to hide. Overall, this provides a reactive method to judge and change the helicopter's visibility, which is easy to deploy and especially suitable for implementation in complex unknown environments.

### 4.2. Finite State Machine

Visibility judgement and target and threat recognition are fundamental factors of decision making in low-altitude flight. The visual servo control and terrain avoidance methods presented in this research were verified to be effective in target tracking and obstacle avoidance. On this basis, a finite state machine was established to combine the decision-making and control methods, thus forming the overall framework for unmanned helicopter operations in low-altitude flight, as shown in Figure 19. The finite state machine established a continuous operation process without human interference and covered most scenes in the low-altitude flight.

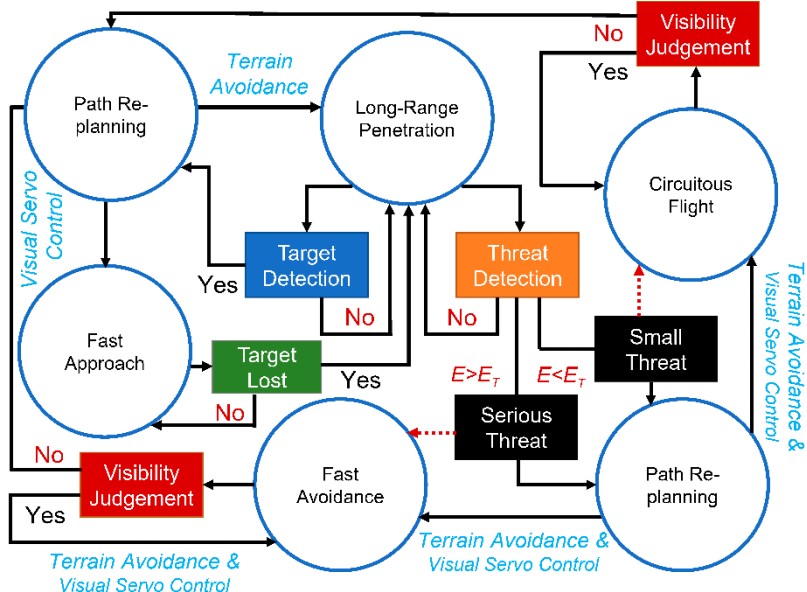

**Figure 19.** Finite state machine of the decision-making framework.

In low-altitude flight, a helicopter is given a distant destination and required to approach the destination at low altitude. Meanwhile, the detection network works to detect targets and threat facilities. Once the target is detected, the helicopter immediately heads to the target through visual servo control and revises the target's position. In this case, the helicopter does not know its exact position with respect to the target on the map; however, it is able to estimate the distance and yaw direction using airborne equipment. Therefore, the helicopter can estimate the target's direction on the basis of its heading. Several target points are placed along the target direction for path replanning of the VFH method. The helicopter continues heading toward the target. Generally, keeping the target in the center of the camera can ensure that the target is not lost during the flight approach. In case the target is lost, the helicopter continues flying to the defined target points to approach the target and returns to visual servo control when the target is rediscovered. After arriving at the target, the helicopter can revise the destination location to fly to the original destination or end the flight mission. This fast approach process can realize a quick attack on or reconnaissance of specific facilities.

If the detection network detects a threat during the flight, the threat degree $E$ is firstly evaluated. We propose a simple way to quantitatively evaluate the threat, as expressed in Equation (22). $\xi_{class}$ is the coefficient for different classes of threats. $S_{box}$ implies the distance to the threat. Basically, a closer distance indicates a greater threat. There are more intricate methods of threat evaluation, but they are outside the scope of this research. We calculated the threat degree mainly to distinguish between serious threats and small threats, so as to design different control strategies. We defined a threat threshold $E_T$, whereby a threat degree that higher than $E_T$ is considered as a serious threat. In this situation, the helicopter executes fast avoidance flight to escape the sight range of the threat as soon as possible, seeking terrains as cover to change its visibility. For this purpose, the helicopter is forced to head to the threat for visibility judgement. The target points of the VFH method are reset as the points of the history path. Then, the helicopter executes lateral maneuvers so as to quickly restore invisibility behind terrain cover. When the threat is lost in the camera, the helicopter can fly away from the threat or carry out further orders. In this research, we gave a higher priority to fast avoidance than to fast approach. The helicopter executes fast avoidance upon detecting a serious threat, regardless of whether a target is detected in the view. If multiple threats are detected during the flight, the helicopter heads toward the threat with highest threat degree to execute fast avoidance.

If the threat degree is lower than $E_T$, the detected threat is considered a small threat, and the helicopter executes circuitous flight. In this situation, the top priority is still restoring the helicopter's invisibility, which requires the helicopter to head toward the threat and seek cover. However, the helicopter is allowed to move laterally. The target points of the VFH method are reset to the sides of the helicopter. When the threat is lost, the helicopter resets the target points and continues its approach to the destination. In this study, the helicopter automatically placed the target points on the opposite site of the threat and followed the path of the VFH method. Many other path-planning methods can also be used for circuitous flight; however, we mainly focused on introducing visibility judgement and realizing reactive threat avoidance.

$$E = S_{box}\xi_{class}. \tag{22}$$

The finite state machine presents a detailed decision-making framework through the state transitions of different flight tasks, such as long-range penetration, fast approach, fast avoidance, and circuitous flight. These flight tasks contain multiple objectives, and their implementation is complicated. Here, we designed the control law for each flight task, all the control methods were derived from the visual servo control and terrain avoidance method in Section 3. Since the control channels are decoupled as illustrated in Section 2.2, we can clearly explain the control method through the control commands of different channels.

For the long-range penetration task, the helicopter approaches the destination and avoids terrain obstacles according to the VFH method. Specifically, the VFH method provides the control commands of the yaw channel and altitude channel, as shown in Equation (23). $\boldsymbol{R}_{GB}$ is the conversion matrix from ground coordinates to body coordinates. The desired direction produced by the VFH method is defined in the ground coordinates and must be converted to body coordinates to generate control commands. $\overline{u}$, $\overline{v}$, and $\overline{w}$ are the control gains of the linear velocities. The longitude channel and lateral channel maintain the helicopter's stability.

$$\begin{cases} \psi = \psi_{VFH} \\ u = \overline{u} \\ v = 0 \\ w = \overline{w}\boldsymbol{R}_{GB}Z_{VFH} \end{cases}. \tag{23}$$

For the fast approach task, the control command of the helicopter yaw channel is provided by the visual servo control method allowing the helicopter toward head to the target. The target points are placed according to the heading direction of the helicopter, as shown in Equation (24), where $d$ is a predefined value that affects the interval between target points, and $\overline{Z}$ is the reference altitude of the low-altitude flight. The longitude channel, lateral channel, and altitude channel are all controlled by the VFH method, as shown in Equation (25), enabling safe terrain avoidance when the helicopter heading direction is locked to the target. If the target is lost in view, which may be caused by detection network failure, helicopter attitude oscillation, or terrain occlusion, the fast approach task is converted back to the long-range penetration task with revised target points.

$$\begin{cases} X_{Target} = X + nd\cos\psi \\ Y_{Target} = Y + nd\sin\psi \\ Z_{Target} = \overline{Z} \end{cases}, n = 1, 2, 3, \ldots \tag{24}$$

$$\begin{cases} r_{Ref} = -k_r(b_x - b_{mid}) \\ u = \overline{u}\boldsymbol{R}_{GB}X_{VFH} \\ v = \overline{v}\boldsymbol{R}_{GB}Y_{VFH} \\ w = \overline{w}\boldsymbol{R}_{GB}Z_{VFH} \end{cases}. \tag{25}$$

The control laws of the fast avoidance and circuitous flight are similar to those of the fast approach, as shown in Equation (23). The VFH method provides different maneuvers

by setting different target points. The target points of fast avoidance and circuitous flight are placed as shown in Equations (26) and (27), respectively, where $T$ is the predefined time interval to sample the history path points, and ($X_{Destination}$, $Y_{Destination}$) are the coordinates of the original destination. For circuitous flight, the target points are placed on the side of the helicopter, and the desired direction of the VFH method is converted to lateral control commands in the body coordinates. The direction of the lateral maneuver is determined according to the threat position, destination position, and history path. Generally, using three control channels to follow the control commands of the VFH method can ensure that the helicopter safely escapes the threat and takes cover behind the terrain to change its visibility.

$$
\begin{cases}
X_{Target} = X_{t-nT} \\
Y_{Target} = Y_{t-nT} \quad , n = 1, 2, 3, \ldots \\
Z_{Target} = Z_{t-nT}
\end{cases}
\tag{26}
$$

$$
\begin{cases}
X_{Target} = X - nd\sin\psi\,\mathrm{sgn}(\Delta(\begin{bmatrix} \cos\psi \\ \sin\psi \end{bmatrix}, \begin{bmatrix} X_{Destination} - X \\ Y_{Destination} - Y \end{bmatrix})) \\
Y_{Target} = Y + nd\cos\psi\,\mathrm{sgn}(\Delta(\begin{bmatrix} \cos\psi \\ \sin\psi \end{bmatrix}, \begin{bmatrix} X_{Destination} - X \\ Y_{Destination} - Y \end{bmatrix})) \quad , n = 1, 2, 3, \ldots \\
Z_{Target} = \overline{Z}
\end{cases}
\tag{27}
$$

## 5. Simulation Experiments

In order to verify the performance of the proposed control framework, we built a typical mountainous map including target or threat facilities, as shown in Figure 20. The target and threat facilities were set at the same position, as were the helicopter initial position and original destination, to better compare the flight performance of the different flight tasks. We designed four flight scenes by defining different facilities in the target/threat position to carry out long-range penetration, fast approach, fast avoidance, and circuitous flight. The installation and parameters of the virtual camera and Lidar were the same as those of the simulations in Section 3.

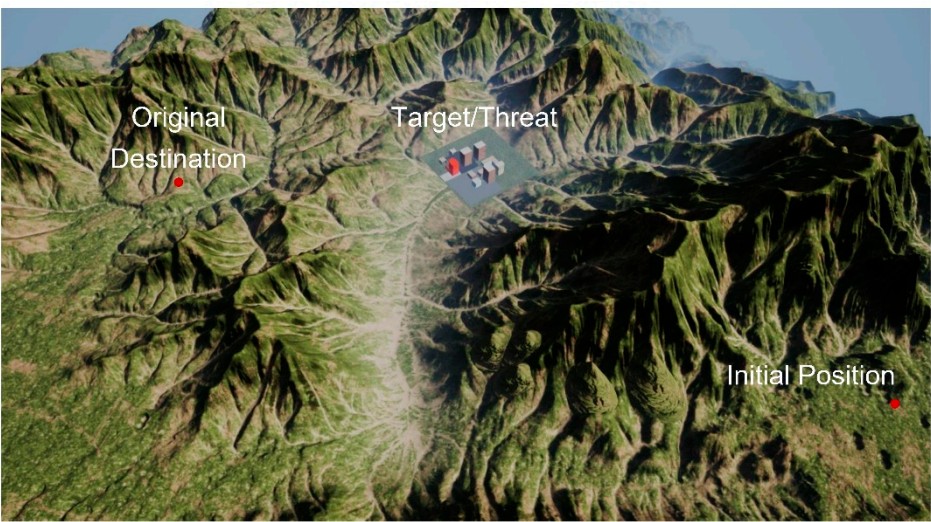

**Figure 20.** Simulation scene of the low-altitude flight.

The long-range penetration flight path is illustrated in Figure 21a. The target/threat was removed from the map. The helicopter followed the command of the VFH method on the premise of low altitude. The helicopter first flew along the hillside of the right-side mountain, and then selected the middle valley to approach the destination. At this moment, the left-side terrain blocked the destination, and the helicopter continued to fly along the hillside of the left-side terrain. When the terrain altitude became low, the helicopter went

over the terrain gap and finally arrived at the destination. The whole flight path was similar to that observed for terrain avoidance in Section 3.2, showing good performance.

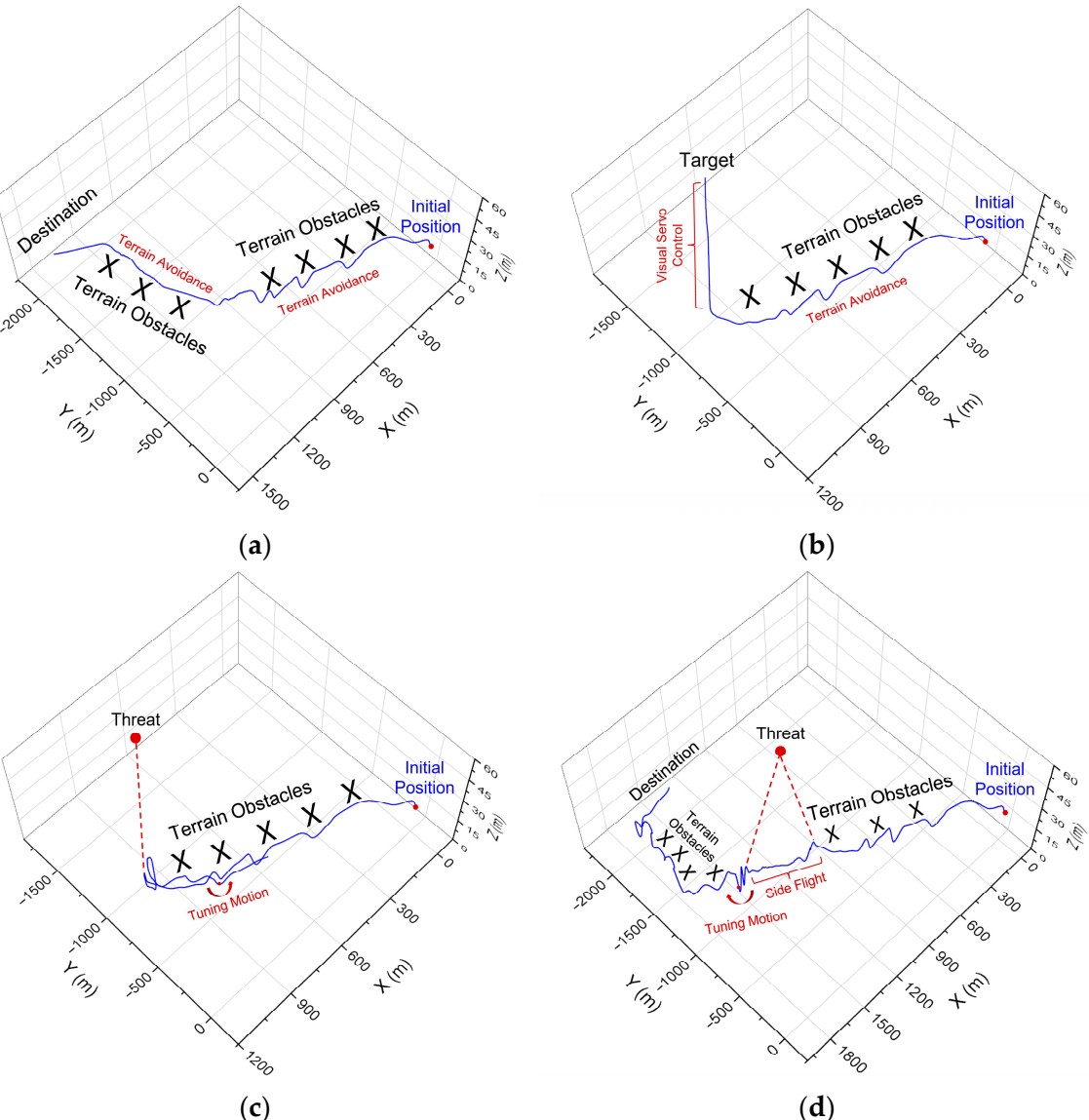

**Figure 21.** Flight paths of the flight tasks in the low-altitude flight: (**a**) long-range penetration; (**b**) fast approach; (**c**) fast avoidance; (**d**) circuitous flight.

The flight path of the fast approach is illustrated in Figure 21b. A target facility was set in the target/threat position. At the initial position, the target was blocked by the right-side mountain, and the helicopter followed the command of the VFH method. When the helicopter bypassed the right-side mountain, the target was detected. The helicopter flew straight toward the target along the middle valley. Here, the forward velocity could be tuned by changing the control gain $\bar{u}$ to realize a faster approaching speed while satisfying other task requirements. The sight range of the middle valley was wide; the helicopter kept the target in view and finally reached the target position.

Figure 21c shows the flight path of the fast avoidance task. The facility in the target/threat position was identified as a serious threat by the detection network. When the threat was detected, the target points of the VFH method were reset as the path points behind the right-side mountain. The helicopter maintained its heading toward the threat and then moved horizontally to fly away from the threat. The control gains of the linear velocities could be tuned to decrease the flight oscillations and increase the flight stability.

Finally, the helicopter flew behind the obstacles and changed its visibility in a short time, verifying the effectiveness of the fast avoidance method. The helicopter continued to fly laterally for a while after the threat was lost from view, before turning around to head to the target points.

Figure 21d shows the flight path of the circuitous flight, which mainly contained three flight phases. The facility in the target/threat position was identified as a small threat. Firstly, the helicopter executed long-range penetration and flew along the hillside of the right-side mountain, just as in the previous flight simulations. Then, the helicopter detected the threat and carried out visual servo control to maintain its heading toward the threat. The target points of the VFH method were reset and placed on the left side of the helicopter, which was closer to the destination. The helicopter moved laterally to the left side and finally flew behind the left-side terrain, changing its visibility. Subsequently, the target point was set as the original destination, and the helicopter turned left to approach the destination. Here, the VFH method considered the cost of the current direction, and we could change the path weight to ensure that the helicopter would not return to the threat once its visibility changed. We could also manually define a rule for the helicopter to choose a direction away from the threat. If the threat was detected again during the flight, the helicopter would repeat the above operations to change the target points and escape the threat. In the flight simulation, the helicopter bypassed the left-side terrain and finally arrived at the destination without detecting the threat. For a better comparison of the flight paths, we present all flight tasks in Figure 22. The decision making and approximate paths of all flight tasks are shown, verifying the overall control framework and control method proposed in this study.

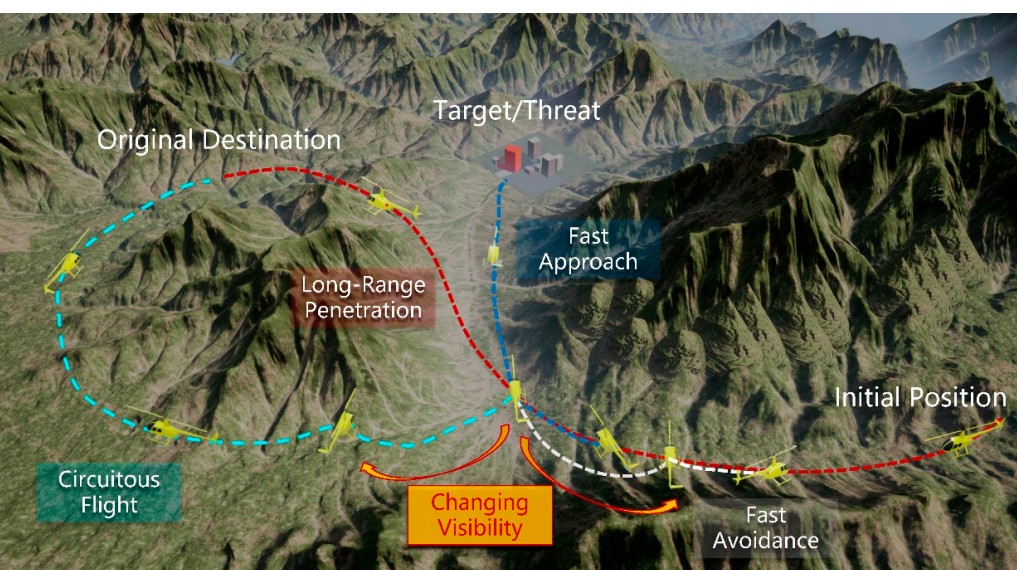

**Figure 22.** Diagrammatic presentation of all flight tasks.

## 6. Conclusions

In this study, the implementation of unmanned helicopter operations for low-altitude flight was investigated. Specific flight scenes in mountainous terrains were discussed in detail. We introduced target and threat recognition into the overall control framework, and we disassembled the low-altitude flight into several basic tasks. The target and threat were identified using the YOLO network. Using the anchor box of the YOLO network, the helicopter realized stable and effective visual servo control in the flight simulations. The 3D VFH method was used for terrain avoidance of the helicopter, achieving good adaptability and performance in unknown mountainous terrains.

Visibility judgment is crucial for low-altitude flight, yet it was rarely investigated in previous research. We proposed a direct-viewing method which can quickly estimate

helicopter visibility without comprehensive map information or threat positions. On this basis, we built the overall control framework using a finite state machine. Such a design incorporated four flight tasks to cover most flight scenes encountered in low-altitude flight. A coupling control method of visual servo control and terrain avoidance was developed to realize these tasks, and their performance was verified through high-fidelity flight simulations.

Using the overall control framework presented in this research, the helicopter could automatically complete complex flight tasks such as fast attack, cover concealment, and circuitous flight similar to human pilots. The control methods are explicable, and the control gains can be tuned to adapt to various flight tasks and scenes. Furthermore, some implementations of the framework can be optimized. For example, the detection network may lose the target, rendering the visual servo control invalid. This can be improved by designing state observers and filters. The target point selection of the VFH method can be further optimized to improve the flight performance and efficiency.

**Author Contributions:** Conceptualization, Z.J., L.N., D.L. and J.X.; methodology, Z.J., L.N. and D.L.; software, Z.J. and L.N.; validation, Z.J. and L.N.; formal analysis, Z.T.; investigation, Z.J.; resources, D.L. and J.X.; data curation, Z.J.; writing—original draft preparation, Z.J. and L.N.; writing—review and editing, D.L. and Z.T.; supervision, D.L. and Z.T.; project administration, D.L. All authors have read and agreed to the published version of the manuscript.

**Funding:** This research received no external funding.

**Institutional Review Board Statement:** Not applicable.

**Informed Consent Statement:** Not applicable.

**Data Availability Statement:** Not applicable.

**Conflicts of Interest:** The authors declare no conflict of interest.

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
