# Peer review of "An Autonomous Control Framework of Unmanned Helicopter Operations for Low-Altitude Flight in Mountainous Terrains"

_drones, doi:10.3390/drones6060150_

Round 1

Reviewer 1 Report

Comments to the authors

Manuscript ID: drones-1776190

Title: An Autonomous Control Framework of Unmanned Helicopter Operations for Low-altitude Flight in Mountainous Terrains

1) Provide a list of contributions at the end of Section 1. Also, consider providing sufficient evidences to support your claim.

2) Avoid defining an acronym to refer to two different concepts. “TA” refers to Terrain Avoidance and Threaten Avoidance at the same time.

3) In essence, the problem is similar to the problem of autonomous controlling unmanned vehicles in the presence of obstacles. There are several methods presented in the literature that address that problem. The authors need to enhance the literature review by discussing more relevant publications. For instance, consider discussing the fooling papers to improve the literature review.

[R1]. “UAV environmental perception and autonomous obstacle avoidance: A deep learning and depth camera combined solution”, 2020. [https://doi.org/10.1016/j.compag.2020.105523]

[R2]. “Constrained Control of UAVs in Geofencing Applications”, 2018. [https://doi.org/10.1109/MED.2018.8443035]

[R3]. “UAV Obstacle Avoidance Algorithm to Navigate in Dynamic Building Environments”, 2022. [https://doi.org/10.3390/drones6010016]

4) I understand that proving stability, convergence and other properties of the utilized cascade PID framework as in Figure  1 is out of the scope of the submitted manuscript. However, the authors need to refer the readers to an appropriate source for more details. Such framework has been discussed in [R2] (mentioned above) which can be used to support its utilization.

5) Simulation section could be improved. In particular, the study lacks a proper comparison with a state-of-the-art method. This would help the readers to understand pros and cons of the proposed method.

Reviewer 2 Report

The research is simulation-based case of study on the problem of autonomous flight of unmanned helicopter on low altitude. In my opinion the paper shed a new light to well-known problem.

The topic is presented clear, but in opinion the Authors should:

- present more data about the simulation environment which was used in particular experiments,
- improve the figures quality - avoid of using pixels, try with vector graphix,
- rearange the references - a lot number is quite old, and there are new publications on this topic coming still.

Round 2

Reviewer 1 Report

No further comment. Thank you for the answers.